# MIX-NET: Deep Learning-Based Point Cloud Processing Method for Segmentation and Occlusion Leaf Restoration of Seedlings

**DOI:** 10.3390/plants11233342

**Published:** 2022-12-01

**Authors:** Binbin Han, Yaqin Li, Zhilong Bie, Chengli Peng, Yuan Huang, Shengyong Xu

**Affiliations:** 1College of Engineering, Huazhong Agricultural University, Wuhan 430070, China; 2Key Laboratory of Agricultural Equipment for the Middle and Lower Reaches of the Yangtze River, Ministry of Agriculture, Wuhan 430070, China; 3Shenzhen Institute of Nutrition and Health, Huazhong Agricultural University, Shenzhen 518000, China; 4Shenzhen Branch, Guangdong Laboratory for Lingnan Modern Agriculture, Genome Analysis Laboratory of the Ministry of Agriculture, Agricultural Genomics Institute at Shenzhen, Chinese Academy of Agricultural Sciences, Shenzhen 518000, China; 5School of Mathematics and Computer Science, Wuhan Polytechnic University, Wuhan 430023, China; 6College of Horticulture and Forestry Sciences, Huazhong Agricultural University, Wuhan 430070, China; 7Key Laboratory of Horticultural Plant Biology, Ministry of Education, Wuhan 430070, China; 8Electronic Information School, Wuhan University, Wuhan 430072, China

**Keywords:** point cloud segmentation, point cloud completion, leaf area measurement, MIX-Net, seedlings, deep learning

## Abstract

In this paper, a novel point cloud segmentation and completion framework is proposed to achieve high-quality leaf area measurement of melon seedlings. In particular, the input of our algorithm is the point cloud data collected by an Azure Kinect camera from the top view of the seedlings, and our method can enhance measurement accuracy from two aspects based on the acquired data. On the one hand, we propose a neighborhood space-constrained method to effectively filter out the hover points and outlier noise of the point cloud, which can enhance the quality of the point cloud data significantly. On the other hand, by leveraging the purely linear mixer mechanism, a new network named MIX-Net is developed to achieve segmentation and completion of the point cloud simultaneously. Different from previous methods that separate these two tasks, the proposed network can better balance these two tasks in a more definite and effective way, leading to satisfactory performance on these two tasks. The experimental results prove that our methods can outperform other competitors and provide more accurate measurement results. Specifically, for the seedling segmentation task, our method can obtain a 3.1% and 1.7% performance gain compared with PointNet++ and DGCNN, respectively. Meanwhile, the $R^{2}$ of leaf area measurement improved from 0.87 to 0.93 and MSE decreased from 2.64 to 2.26 after leaf shading completion.

## 1. Introduction

Phenomics is a discipline that studies the observable morphological characteristics and their change patterns exhibited by individual plants or groups under specific conditions [1]. Plant phenomics is a key technology to further explore the intrinsic genotype–phenotype–environment association, and provides technical support for genomic functional analysis, molecular breeding, and precise management of agricultural production [2,3]. However, for plants, leaves are the most important for their external morphology and physiological functions [4]. Most of the traditional leaf measurement methods use the two-dimensional projection of the leaf on the CCD plane in a two-dimensional image to generate pixel points, so as to calculate the leaf parameters [5,6]. In practice, measurements based on 2D images cause serious measurement errors because the growth pattern and natural deformation of the plant make it impossible for the leaf to be an absolute plane. To solve this problem and achieve more accurate measurements, it is crucial to capture the morphological configuration of leaves in three dimensions. With the rapid development of sensing technology and the improvement of computational performance, we can easily accomplish rapid data acquisition and phenotype extraction on a 3D scale. For example, LiDARR [7], low-cost RGB-D depth cameras [8], and multi-view imaging techniques [9] have been widely used in 3D plant data acquisition. However, the current techniques for processing 3D plant data, LiDAR and multi-view imaging, are very time-consuming and require much manual intervention, resulting in the accumulation of large amounts of raw data [10]. In addition, these methods limit the high throughput resolution of phenotypic indicators of interest to agronomists. However, low-cost RGB-D depth cameras are widely used in mapping, 3D reconstruction, indoor robotics, gesture recognition, and object detection and recognition due to their low cost, high measurement accuracy, and fast measurement speed [11]. Nowadays, low-cost RGB-D depth cameras are also increasingly used in plant phenotyping techniques.

In the process of measuring plant phenotypes with low-cost RGB-D depth cameras, one method is to measure plant phenotypes in a complete 3D point cloud, mainly through multi-view 3D point cloud registration [12]. This method can completely eliminate occlusions and obtain high-accuracy point clouds, but the point cloud alignment is demanding and time-consuming for image processing algorithms. Moreover, there is no good solution to the flexible registration problem under the jitter condition. Another approach is to measure plant phenotypes mainly in single-view 3D point clouds by using the mapping relationship between color and depth images [13]. However, this method is a single-view 3D point cloud, and the occlusion and overlap between leaves becomes more and more serious as the plant grows during the measurement process, which leads to more and more errors in measuring phenotypes. Especially at present, with the increasing demand for high-throughput acquisition, a low-cost, accurate measurement, and high-throughput method is urgently needed. So this paper proposes a 3D leaf shading segmentation and restoration technique using a low-cost RGB-D depth camera in a single view. This technique overcomes the problem of inter-leaf occlusion and overlap in current depth camera measurements of plant phenotypes to increase the accuracy of plant phenotype measurements over longer growth periods. However, there are still many issues to be addressed in order to achieve this goal.

First is the need to address the quality of point cloud data. Specifically, the data obtained from RGB-D sensors are coarse, and outlier and hover point noisy point clouds are severe [14]. Traditional methods use radius filtering and straight-pass filtering to filter out these noisy point clouds. For example, the researchers used Kinect V2 to capture rapeseed point clouds and remove other outliers and hover point noise based on line of sight and surface normal [15]. Although it can effectively improve the quality of the point cloud, it will remove some important points. A depth image-based filtering approach has also been proposed that can filter the noise of hover points and outliers well, but this approach cannot be directly applied to 3D point clouds [12]. Considering the shortcomings of these methods, hover points are generated because the light source is refracted when injected into the edge of the object, resulting in the receiver not receiving the signal properly. Therefore, this paper is mainly based on the fact that the normal vector of normal points in the depth camera when photographing the object has a very different offset angle with respect to the camera coordinate system line of sight compared with the hover point. As well as the sparse characteristics of outlier points, the two are combined to determine the offset angle of each normal vector relative to the camera line of sight and the spatial density of the point cloud. A new method based on domain space constraint is proposed to greatly filter out the hover points and outlier noise in 3D space.

Secondly, in the process of measuring plant phenotype, we need accurate leaf segmentation results. Especially under the condition of single view cloud processing, the phenotype detection accuracy is more sensitive to the segmentation accuracy. Due to the mapping between the color image and the depth image, depth cameras often segment in the color image and then align with the depth image to obtain the target segmentation result [16]. However, the resolution of color images of mainstream depth cameras such as Azure Kinect is much higher than that of depth images, which makes the quality of point clouds converted by this alignment effect poor. Moreover, considering the single-view object, although occluded and overlapped in the two-dimensional image, there is a certain spatial distance in the three-dimensional point cloud. So in this paper, we mainly focus on the segmentation of single-view 3D point clouds captured by depth cameras. In recent years, phenotypic measurements based on 3D models have attracted more and more research [17,18,19,20]. For example, an octree algorithm has been used to divide a single-plant point cloud into many small parts, and then each part is combined into one organ for segmentation based on the spatial topology [21]. Others first segmented 2D images using distance transform and watershed algorithms, and then performed leaf segmentation before mapping the segmented images into 3D [22]. However, these aforementioned methods require a lot of human interaction, rely on empirical parameter settings, and cannot meet the requirements of high-throughput processing in plant phenotyping studies. In contrast, deep learning-based methods can automatically extract features from large data volumes through algorithm design, providing a new perspective to address these issues [23]. For example, semantic segmentation of tomato plants in greenhouses was developed using PointNet++ [24] and further estimated leaf area indices [25]. A point cloud grid segmentation algorithm has also been improved and a hybrid segmentation model has been proposed that can adapt to the morphological differences of different individuals of cotton to achieve stem and leaf separation [5]. Recently, a point cloud segmentation network with dual-neighborhood feature extraction and dual-granularity feature fusion has been proposed to achieve semantic segmentation and leaf instance segmentation of three plants simultaneously [26]. However, the experimental materials of these methods mentioned above are mostly generated by LiDAR and multi-view 3D reconstruction techniques with good point cloud quality and less occlusion between plant leaves. However, the experimental accuracy of these methods drops dramatically when dealing with single-view 3D point clouds from RGB-D depth cameras due to the presence of large amounts of occlusion and overlap.

We consider that it is because most of the feature extraction of the above methods only divide the point cloud into local point cloud blocks to enrich the extraction of local features of the point cloud. However, for the interaction between point cloud blocks, the above methods all only perform simple interaction between adjacent blocks. Subsequently, some people also use attention mechanisms to enhance this interaction, but this approach will occupy a lot of space, and there is no simple and effective attention mechanism for 3D point clouds that can well solve the segmentation of such overlapping point clouds. Therefore, we develop a point cloud segmentation method based on the U-net shape hybrid point-mixer mechanism, referred to as MIX-Net, which consists of a continuous encoder–decoder network. In the encoder, considering previous point cloud feature extraction methods, such as PointNet++ [24] and DGCNN [27], the point cloud is first transformed into point cloud blocks using the K-nearest neighbor algorithm, and the features of each point cloud block are extracted by a convolutional neural network (CNN) or a custom convolutional module. However, converting the whole point cloud into point cloud blocks is too costly, there is also a large amount of redundancy, and the interaction between point cloud blocks is missing. We propose a simpler interactive feature fusion module that samples key points in the complete point cloud using point cloud curvature sampling, and then uses the key points for K-nearest neighbors to form point cloud blocks, which reduces the computational cost and redundancy between point cloud blocks. In addition, to increase the feature interaction between point cloud blocks, we borrow the success achieved by purely linear mlp-mixer in 2D images [28]. We design a purely linear point-mixer feature interaction network. It mainly includes the feature interaction within the interactive feature fusion point cloud blocks and the feature interaction between each point cloud block. Finally, considering that this interactive feature fusion module loses some feature information, we also use multi-resolution feature extraction to extract the deep features of the missing point clouds. In the decoder, we employ the up-sampling module [29] to incrementally generate higher resolution point clouds, and after up-sampling, we resort to the point-mixer feature interaction network to generate complete point clouds with detailed details. Finally, we not only segmented the plant stems and leaves, but also added the mean-shift clustering algorithm at the end of the network to segment the instances among the leaves. Through experimental comparison, we found that this feature extraction and interaction approach achieves good results when dealing with plants with occlusions and overlaps.

A final challenge is the completion of missing leaves in segmented plant leaves. Although this technique is an emerging field in the application of 3D phenotypes to plants, it has been an important research problem in the graphics and vision community. For example, Poisson reconstruction has been used to complete holes in the surface of objects, but this method is characterized by a small patching area [30]. The geometric symmetry of the object is used to complete the complete object, but this method is characterized by a low quality of completion [31]. The above traditional methods perform poorly in the plant-completion task because they can only handle simple missing data and are less effective for missing plant leaves due to their varying angles and degrees of missingness. In recent years, 3D point cloud completion methods based on deep learning have achieved great success, which provides insights to solve the plant data problem. For example, a voxel-based grid algorithm has been developed to repair incomplete input data. However, the voxel-based approach is limited by its resolution, as the computational cost increases significantly with the resolution [32]. There are also global features learned first from a partial input point cloud to produce a rough complete point cloud and generate more details by collapsing the decoder operation [33]. Recently, researchers have proposed a point cloud fractal network for repairing incomplete point clouds using partial point clouds as input to keep the space of the original part unchanged and output only the missing part of the point cloud instead of the whole object [34]. Although the deep learning-based point cloud completion method has made some research progress, it still faces some challenges such as large computation and low resolution, and it is difficult to cope with the missing leaves in plants due to various missing angles and different degrees of missingness. Therefore, we would like to propose a new method that is efficient, stable, and applicable to plant leaf restoration. Firstly, for various missing angles and different degrees of missing in plants, we adopt a self-supervised learning training approach by using existing intact leaves, setting 14 missing angle viewpoints (8 vertices of squares and 6 face centers of squares) in 3D space by 3D squares, finding the distance from the viewpoints to the leaves, and removing the distance from the viewpoints to the leaves by different degrees of the missing set (15%, 25%, 50%). The nearest distance from the viewpoint to the leaf is removed to generate the missing leaf data. The network structure, we use the same network structure as the plant leaf segmentation, only in the last layer of the network structure and the loss function is different. We found that MIX-Net can also be well adapted to the plant leaf completion task and achieved good experimental results.

In conclusion, the current research methods cannot achieve satisfactory results in the tasks of denoising, plant leaf segmentation, and plant leaf completion for single-view 3D point clouds of depth cameras. To this end, this paper uses seedlings as the experimental object and first proposes a neighborhood spatial constraint method using a combination of spatial density of point clouds and differences in the angle of normal vectors relative to the camera view offset. In the filtering process of the seedling point cloud, not only can the hover points and outliers around the seedling point cloud be filtered out, but also relatively small details, such as relatively thin and narrow stems, can be retained. Subsequently, we also developed a plant point cloud segmentation and plant leaf completion method based on a U-net shape hybrid point-mixer mechanism, referred to as MIX-Net. This method consists of two main components: (1) a neighborhood aggregation strategy, which mainly transforms a complete point cloud into a sequence of point cloud blocks; (2) a point-mixer mechanism, which allows for enrichment within and between point cloud blocks. To demonstrate the effectiveness of our method, we constructed a dataset of single-view seedling point clouds containing real labels for seedling segmentation and seedling leaf completion tasks. Experimental results show that the method not only balances the two tasks of plant segmentation and plant leaf completion well, but also both tasks obtain satisfactory performance on various common datasets. Moreover, in experiments on seedlings with occlusion and overlap, the method was able to separate stems and leaves under occlusion and to complete the missing leaves. In summary, our method provides a critical solution for inter-leaf shading and overlap in depth camera-based plant phenotyping studies. It makes it possible to achieve high-throughput acquisition and high-precision measurement of plant phenotypes using depth cameras.

## 2. Materials and Methods

### 2.1. Experimental Materials

The experimental subjects of this paper were typical melon seedlings, including watermelon seedlings (zaojia 8424), pumpkin seedlings (Jingle Fengjia), and cucumber seedlings (Jinchun No. 2). Samples were grown in the greenhouse of the Central China Branch of the Vegetable Crop Improvement Center of Huazhong Agricultural University from January 2021 to March 2021, and phenotypes were determined at the Key Laboratory of the Ministry of Horticultural Plant Biology. Seedlings were soaked in warm water, removed and drained, wrapped in gauze, and placed in a 28 °C thermostat for germination, and sown in 50-hole cavity trays. The mass ratio of grass charcoal, vermiculite, and perlite in the seedling substrate was 3:1:1, and Yara miaole compound fertilizer (1.0 kg/m${}^{3}$) was added to the substrate before sowing. After that, seedlings were cultivated in an artificial climate chamber at a diurnal temperature of 28 °C–18 °C and humidity of 65–85%, and seedlings were sprayed 1000 times with lairui Seedling Compound Fertilizer No. 1 after the 1-leaf-1 stages and 800 times with Yara Seedling Compound Fertilizer No. 1 until the end of 3-leaf-1 stages.

### 2.2. Mix-Net Based Seedling Point Cloud Processing Method

#### 2.2.1. Overview

Figure 1 shows the flow of the method in this paper with watermelon seedlings as an example. The method consists of five main parts: high-throughput data acquisition of seedlings, point cloud preprocessing, datasets construction, point cloud segmentation and completion, and leaf area calculation.

#### 2.2.2. Data Acquisition

Five trays (160 plants in total) of watermelon seedlings, cucumber seedlings, and pumpkin seedlings were grown for algorithm design and validation experiments using the method in Section 2.1, and destructive experiments were taken at the 1-leaf-1 stages, 2-leaf-1 stages, and for 3-leaf-1 stages phenotypic algorithm validation. High-throughput data acquisition was performed by using a semi-automatic image acquisition platform for single seedlings. It can be observed that a depth camera is mounted directly above the seedlings, and the depth camera is connected to an external computer. Target plant seedlings grown in the intelligent greenhouse are transplanted into pots and then placed in the instrument where Azure Kinect is deployed. Image acquisition and processing algorithms were developed using Azure Kinect SDK 1.4.1, Microsoft Visual Studio 2019, Window10 OS, and Tesla P100 GPU. Using this software system, 1024 × 1024 depth images can be acquired. Image acquisition was performed in a room with natural light. For the seedlings after image capture, the hand-picked flattened leaves were scanned using an Epson Expression 12000XL scanner to obtain leaf area data.

#### 2.2.3. Point Cloud Preprocessing

The point clouds collected by the Azure Kinect platform are dense, containing approximately 600,000 to 1 million points per plant. However, it contains a large amount of background noise. Hence, to ensure the accuracy and integrity of the data, the original point cloud needs to be processed by background removal and point cloud filtering before being used in subsequent steps. As shown in Figure 1b, the processing steps are as follows.

(1) Straight-pass filtering to filter out the background. For the invalid background beyond the seedlings, thresholds of 0.5 m, 0.5 m, and 0.7 m in length, width, and height are selected for direct-pass filtering since the camera height is known. Subsequently, flat ground is fitted using least squares to split the ground from the seedlings, thus removing the ground. Finally, for the seedling tray, the ground is advanced in the opposite direction of the z-axis by 0.11–0.13 as the cutting point, and the seedling tray is removed using direct-pass filtering on the z-axis.

(2) Point cloud filtering is based on the neighborhood space constraint. It includes the following steps.

The original point cloud is filtered by (1) to obtain the point cloud containing only the plant area.Set a threshold N, find N neighborhoods around each centroid using KNN, and find the average value D of the Euclidean distance between the centroid and the neighborhoods.The angle W between the normal vector and z-axis is solved by fitting the plane with least squares to predict the normal vector of each centroid through the set neighborhood threshold N.Repeat the above operations First and second, if D ≥ d or W ≥ c, it is judged to be a hover point, and the point is deleted. Iterate through the whole point cloud to eliminate all the hover points.

Through comparison experiments, it is found that the best filtering effect is achieved when the parameter N is 12, d is 0.0034 and c is 60°. The setting of parameters is entirely based on the adjustment of the algorithm, independent of the parameters of the camera and the external shooting environment of the plant. Compared with the traditional point cloud filtering method, this method can not only remove the suspended points well, but also retain a large amount of point cloud details.

#### 2.2.4. Datasets Construction

Point cloud data were collected from 50 samples of each of three types of melon seedlings (watermelon seedlings, cucumber seedlings, and pumpkin seedlings), covering the 1-leaf-1 stages, 2-leaf-1 stages, and 3-leaf-1 stages of the seedlings. The point clouds were then annotated using Cloud Compare software, and the annotation enhanced the data to four times the original size. A total of 600 seedling point clouds were obtained for the three types of seedlings, which constituted a point cloud segmentation data set. The complete leaf point clouds segmented by Mix-Net were generated by the missing point cloud generation method and formed 1800 point cloud pairs together with the complete point clouds as the point cloud complementary data set, as shown in Figure 1c. The division of the datasets is represented in Table 1.

(1) Data augmentation. Considering that the rotation-translation invariance and scale invariance of the point cloud, the seedling point cloud is subjected to some random panning in [−0.2, 0.2], random anisotropic scaling in [0.67, 1.5] changes to increase the training data.

(2) Point cloud annotation. In this study, cloud compare was used to annotate the training data for stem and leaf segmentation. The stems and leaves of seedlings were first separated by entering the crop tool in the seedling point cloud selection software. For semantic segmentation, manual interactions were given to two different classes of scalar color information. Stem points were marked as 0 and leaf points were marked as 1. For the instance segmentation stem points were marked as 0 and each leaf was marked with a different marker. It takes only about 30 s to mark each seedling point cloud using Cloud Compare software. This is very efficient due to the small size and clear structure of the seedlings.

(3) Missing point cloud generation. Firstly, for various missing angles and different degrees of missingness of plant leaves, we adopt a self-supervised learning training approach by using existing intact leaves, setting 14 missing angle viewpoints (8 square vertices and 6 square face centers) in 3D space by 3D square, finding the distance from the viewpoints to the leaves, and removing the nearest distance from the viewpoints to the leaves by different degrees of missingness set (15%, 25%, 50%) in order to generate missing leaf data. This approach generates missing point clouds similar to the missing occlusion between leaves and allows control of viewpoints as well as radii to simulate more types of missing occlusion. It has been experimentally verified that this missing approach can effectively fill in the occluded leaves.

#### 2.2.5. MIX-NET Network for Segmenting and Completing Point Clouds

**Encoder.** The overall structure of MIX-Net’s encoder is shown in the left half of Figure 2. The aim is to encode the input points into a new high-dimensional feature space. By employing a similar approach to the neighbor point aggregation mentioned in PointNet++ and PointCNN [35], the features of the point cloud are transformed into a new higher dimensional feature space, which characterizes the semantic affinity between neighbor points and serves as the basis for various point cloud processing tasks. The embedded features are then fed into the point-mixer module to learn the rich semantics within each neighbor point and the rich semantic and discriminative representations between individual neighbor points. To obtain richer point cloud features, the encoder uses multi-resolution point cloud feature extraction with 2048, 1024, 512, and 256 resolutions and neighbor points extracted in 32, 16, 8, 4, and point-mixer feature processing module dimensions of 1024, 512, 256, and 128.

**Decoder.** The decoder takes the final feature vector as input and aims to output M × 3 to represent the complete 3D point cloud shape, as shown in the right half of Figure 2. To generate higher quality complete 3D point clouds, based on FPN [36], we propose a complete progressive point cloud generation approach with the idea of generating 3D point clouds progressively from low to high resolution, such that primary, secondary, and detailed points will be predicted from layers of different feature depths. The primary and secondary points will try to match their corresponding feature points, gradually increase the number of points by interpolation up-sampling [29], and generate their high-dimensional feature maps, which will be decoded by the point-mixer module to propagate the overall geometric information to the final detailed points. In the whole process of point cloud complementation, the output point cloud resolutions of the four stages are 256, 512, 1024, and 2048; the dimensions are 1024, 512, 256, and 128.

**Point cloud classification.** We use a classification network using MIX-Net to classify a point cloud P into $N_{C}$ classes of objects. The features map is fed to the classification decoder. It consists of two cascaded feed-forward neural networks LBRs (combining linear, batch norm (BN), and LeakyReLU layers), each with a Dropout probability of 0.5. A linear layer is finally used to predict the final classification. Each category scores $C\in R^{N_{c}}$. The category label of the point cloud is determined as the category with the maximum score.

**Point cloud segmentation.** The segmentation point cloud task is to divide it into several parts. A part label must be predicted for each point. To learn a general model applicable to various objects, we also encode the object category vector and connect it to the features map. The structure of the final output is essentially the same as that of the classification network. Then, the segmentation score $S\in R^{N\times N_{s}}$ for each point of the final output point cloud is predicted. Finally, the label with the maximum score for each point is also identified as the label for that segment. For instance, in segmentation, the features are concatenated and then reduced to five dimensions by a feature dimension module (1D Convolution with LeakyReLU). Then the class of each instance is predicted by the mean-shift clustering algorithm.

**Point cloud completion.** We use the same network architecture as in point cloud segmentation. The difference is that in the process of generating the complete point cloud, the feature map of the same resolution in the encoder is fused with the interpolated feature maps to keep the structure of the input missing point cloud unchanged during the decoding work, after which the point-mixer is used to process the features, and then MLP is used to generate the 3D coordinates of the point cloud for each resolution.

#### 2.2.6. Neighborhood Aggregation Strategy

In most previous works, encoders are mostly used for feature extraction with multi-layer perception (MLP). However, they ignore the local neighborhood information, which is essential in the point cloud structure. We design a neighborhood aggregation strategy to enhance local feature extraction with neighborhood point embedding, as shown in Figure 3. More specifically, assume that the neighbor feature aggregation layer takes a point cloud P with N points and corresponding features $F_{n}$ as input and outputs a sampled point cloud $P_{s}$ with $N_{s}$ points and their corresponding aggregated features $F_{s}$. First, we use the curvature sampling algorithm [37] to down-sample $F_{n}$, and generate features $F_{i}$. Then, with each point in feature $F_{i}$ as the center, find the nearest *k* points in feature $F_{i}$ to form a neighborhood $F_{ik}$. Finally, the output features $F_{s}$ in the way shown in Equation (Equation 1):(1)$$F_{s}=MP\left(LBR\left(\mathrm{concat}\left(F_{i}-F_{ik},RP\left(F_{i},k\right)\right)\right)\right)$$
where MP is the maximum pooling operator and $RP\left(x,k\right)$ is the operator that repeats the *x* vector *k* times to form a matrix. To extract more comprehensive features, multi-resolution point cloud feature extraction is used to extract and fuse their features at different resolutions.

#### 2.2.7. Point-Mixer Mechanism

We propose a feature processing network that can interact within and between localities, called a point-mixer. As shown in Figure 4. The point-mixer generates S sequences of non-overlapping point cloud groups as input after passing through a local neighborhood aggregation strategy, and the point cloud group sequences are linearly projected to the desired dimension using the same projection matrix. Where point-mixer consists of multiple layers of the same size, each layer consists of two MLP blocks [28]. The first layer is the point-mixer intra-group blending MLP: it acts between the interiors of the point-mixer and maps the point-mixer sequences. The second is the inter-group hybrid MLP: it acts between point cloud groups and again maps the mapping back to the same dimensions and representation. Each MLP block contains two fully connected layers and a nonlinear operation is applied to each row of its input data tensor independently. The point-mixer process can be written in the form of Equation (Equation 2):(2)$$\begin{array}{cc} F_{c} & =F_{s}+\mathrm{MLP}\left(\mathrm{LayerNorm}\left(F_{s}\right)\right)\\ F_{o} & =F_{c}+T\left(\mathrm{MLP}\left(T\left(\mathrm{LayerNorm}\left(F_{c}\right)\right)\right)\right) \end{array}$$
where *T* is a flip operation. $F_{c}$ and $F_{o}$ are tunable hidden features in the intra-group blending and inter-group blending MLP, respectively. Note that the dimension selection is independent of the number of input point cloud groups.

#### 2.2.8. Loss Function

**Semantic segmentation.** The softmax cross-entropy function was used as a loss function during training and is shown in Equation (Equation 3):(3)$${\mathrm{Loss}}_{\mathrm{sem}\;}=\sum_{n=1}\left(-y_{n}\times \log\left({\hat{y}}_{n}\right)\right)$$
where *n* is the total number of points in the input point cloud; $y_{n}$ is the ground truth of the multi-level classification corresponding to this point cloud; ${\hat{y}}_{n}$ is the probability of the output of each point cloud category using the softmax function. The specific for ${\hat{y}}_{n}$ mulae for are shown in Equation (Equation 4):(4)$${\hat{y}}_{n}=\frac{e^{Jn}}{\sum_{i}e^{J_{i}}}$$

According to Equation (Equation 4) for the known *n*th input point, the value of $J_{n}$ was calculated using Equation (Equation 5):(5)$$J_{n}=w\times q_{n}$$
where *w* is the weight of the network as a whole, and $q_{n}$ is the input parameter for the *n*th point in the point cloud.

**Instance segmentation.**${Loss}_{\mathrm{ins}}$ is given by Equation (Equation 6):(6)$$\begin{array}{c} {Loss}_{\mathrm{ins}}=L_{s}+L_{\mathrm{reg}}\\ L_{s}=\frac{1}{I}\sum_{i=1}^{I}\frac{1}{N_{i}}\sum_{j=1}^{N_{i}}{\left[\max\left[0,{∥c_{i}-f_{j}∥}_{2}-\delta_{s}\right]\right]}^{2}\\ L_{\mathrm{reg}\;}=\frac{1}{I}\sum_{i=1}^{I}{∥c_{i}∥}_{2} \end{array}$$
where *I* represents the number of instances in the current point cloud batch being processed and $N_{i}$ represents the number of points contained in the *i*-th instance; $c_{i}$ represents the center of the points belonging to the *i*-th instance in the current feature space; and $f_{j}$ represents the feature vector of the point *j* in the current feature space. The parameter $\delta_{s}$ defines a boundary threshold that allows the aggregation of points of the same instance.

**Point cloud completion loss.** The loss measure in the point cloud completion process represents the difference between the true complete point cloud corresponding to the missing point cloud and the predicted point cloud. Fan [38] proposed two alignment-invariant metrics to compare the difference between disordered point clouds, namely Chamfer distance (CD) and bulldozer distance (EMD). Because the bulldozer distance (EMD) occupies more memory and takes longer to calculate, while the Chamfer distance (CD) is more efficient to calculate, this paper chooses the Chamfer distance as the loss function for point cloud completion as follows Equation (Equation 7):(7)$$L_{\mathrm{CD}}\left(S_{1},\;S_{2}\right)=\frac{1}{2}\left(\frac{1}{\left|\;S_{1}\right|}\sum_{x\in S_{1}}\min_{y\in S_{2}}∥x-y∥+\frac{1}{\left|\;S_{2}\right|}\sum_{y\in S_{2}}\min_{x\in S_{1}}∥x-y∥\right)$$

The mean nearest square distance, referred to as the Chamfer distance (CD), between the predicted point cloud $S_{1}$ and the true point cloud $\;S_{2}$ is measured using Equation (Equation 13). The progressive deconvolution completion network is a special progressive deconvolution 3D point cloud completion network in which the complete point cloud is generated in four stages with resolutions. The predicted point cloud outputs of the four stages are denoted by $Y_{1}$, $Y_{2}$, $Y_{3}$, and $Y_{4}$; the true complete point clouds sampled from the true point cloud by IFPS to N/8, N/4, N/2, and N resolutions are denoted by $Y_{\mathrm{gt}}$, $Y_{\mathrm{gt}}^{\prime }$, $Y_{\mathrm{gt}}^{\prime \prime }$, and $Y_{\mathrm{gt}}^{\prime \prime \prime }$. The Chamfer distances (CD) of the four stages are denoted by $d_{{\mathrm{CD}}_{1}}$, $d_{{\mathrm{CD}}_{2}}$, $d_{{\mathrm{CD}}_{3}}$, and $d_{{\mathrm{CD}}_{4}}$. The complete loss function for the training process is shown in Equation (Equation 8):(8)$$L_{\mathrm{com}\;}=d_{{\mathrm{CD}}_{1}}\left(Y_{1},Y_{\mathrm{gt}}\right)+d_{{\mathrm{CD}}_{2}}\left(Y_{2},Y_{\mathrm{gt}}^{\prime }\right)+d_{{\mathrm{CD}}_{3}}\left(Y_{3},Y_{\mathrm{gt}}^{\prime \prime }\right)+d_{{\mathrm{CD}}_{4}}\left(Y_{4},Y_{\mathrm{gt}}^{\prime \prime \prime }\right)$$

## 3. Results

### 3.1. Point-Cloud Noise Removing

The proposed method is compared with statistical filtering, radius filtering, and domain maximum filtering for two-dimensional depth maps. The experimental results are shown in Figure 5b–e, respectively. As one can observe, all three methods result in incomplete filtering if a small radius range is set, as shown in the yellow box in Figure 5b. However, if the radius is large, the other three methods will delete some important points, as shown in the red boxes in Figure 5c,d.

### 3.2. Evaluation Metrics

We compare our MIX-Net with other popular competitors on two public datasets. For a fair comparison we use the same training strategy to optimize our method and competitors and use several popular metrics in point cloud classification and segmentation to evaluate the performance, including accuracy ACC (accuracy) and part-average intersection over union ratio IoU (intersection over union). In the formula, TP (true positives) means the positive class is determined as a positive class, FP (false positives) means the negative class is determined as a positive class, FN (false negatives) means the positive class is determined as a negative class, and TN (true negatives) means the negative class is determined as a negative class.

The accuracy of the *i*-th class of objects in *N* classes is shown in Equation (Equation 9):(9)$$ACC_{i}=\frac{TP_{i}+TN_{i}}{TP_{i}+FP_{i}+FN_{i}+TN_{i}}$$

In the *N* class object, the intersection ratio of the *i* class is shown in Equation (Equation 10):(10)$$IoU_{i}=\frac{TP_{i}}{TP_{i}+FP_{i}+FN_{i}}$$

The average cross-merge ratio of all classes is shown in Equation (Equation 11):(11)$$mIoU=\frac{1}{N}\sum_{i=1}^{N}IoU_{i}$$

The parameter *C* is the number of semantic classes for the calculation of the mean precision (mPrec) and the mean recall (mRec).
(12)$$\begin{array}{cc} \; & \mathrm{mPrec}=\frac{1}{C}\sum_{i=1}^{C}\frac{\left|TP(\mathrm{sem}=i)\right|}{\left|\mathrm{IP}(\mathrm{sem}=i)\right|}\\ \; & \mathrm{mRec}=\frac{1}{C}\sum_{i=1}^{C}\frac{\left|TP(\mathrm{sem}=i)\right|}{\left|\mathrm{IG}(\mathrm{sem}=i)\right|} \end{array}$$

Because the semantic classes include the stem class and the leaf class of each plant species, *C* is fixed at 2. The notation |TP(sem=i)| represents the number of predicted instances whose IoU is above 0.5 in the semantic class *i*. The notation |TP(sem=i)| represents the total number of predicted of instances in semantic class *i*. |IG(sem=i)| represents the number of instances of the ground truth in semantic class *i*.

We evaluate the reconstruction accuracy by calculating the CD between the predicted complete shape and the true shape (Equation (Equation 7)). At the same time, considering the sensitivity of CD to outliers, we also use F-score to evaluate the distance between object surfaces, which is defined as the harmonic mean between precision and recall. EMD is only defined when $S_{1}$ and $S_{2}$ have the same size in Equation (Equation 13):(13)$$L_{EMD}\left(S_{1},S_{2}\right)=\min_{\phi :S_{1}\to S_{2}}\frac{1}{\left|S_{1}\right|}\sum_{x\in S_{1}}{∥x-\phi \left(x\right)∥}_{2}$$
where ϕ is a bijection.

The correlation coefficient ($R^{2}$) and mean square error (MSE) were calculated to compare the results, which can be calculated using Equation (Equation 14):(14)$$\begin{array}{cc} \; & R^{2}=1-\frac{\sum_{l=1}^{m}{\left(v_{l}-v_{l}^{\prime }\right)}^{2}}{\sum_{l=1}^{m}{\left(v_{l}-{\bar{v}}_{l}\right)}^{2}}\\ \; & RMSE=\sqrt{\frac{1}{m}\sum_{l=1}^{m}{\left(v_{l}-v_{l}^{\prime }\right)}^{2}} \end{array}$$
where m denotes the number of objects to be compared; $v_{l}$ indicates the value of the manual measurement result; $v_{l}^{\prime }$ denotes the values of the phenotypic parameters extracted from the segmentation results according to the MIX-Net model; ${\bar{v}}_{l}$ indicates the mean of manual measurement results.

### 3.3. Effectiveness of MIX-Net Network on Seedling Datasets

The experimental performance of MIX-Net on the seedling point cloud datasets was evaluated and compared with other methods in a comprehensive manner. For all network models, the batch size was 32, and each network was trained 250 times individually. The initial learning rate was 0.01, and a cosine function was used to decay the learning rate to adjust the learning rate for each period.

#### Results of Seedling Leaf Segmentation

Experiments were conducted using MIX-Net on the seedling point cloud semantic segmentation datasets and compared with PointNet++ and DGCNN networks. The experimental results are shown in Table 2 below, and MIX-Net improved by 3.1% and 1.7%, respectively. The semantic segmentation example of seedlings segmented by MIX-Net is shown in Figure 6. Additionally, the results of the instance segmentation of seedlings compared with Soft-Group and ASIS are shown in Table 3 and Figure 7 below.

### 3.4. Results of MIX-Net Applied to Leaf Completion under Self-Supervised Learning

We conducted experiments using MIX-Net on the seedling leaf datasets in Section 2.2.4 and compared it with its point cloud completion method. The CD and EMD evaluation metrics are given in Table 4, and the results show that MIX-Net outperforms other networks in both evaluation metrics. The results of leaf completion are given in Figure 8. It can be seen that the completion result ensures the original leaf structure remains unchanged while the output leaf is more uniform.

### 3.5. Results of MIX-Net Applied to Leaf Completion under Supervised Learning

In the above experiments, the training process is self-supervised, but the actual leaf point cloud completion process is supervised learning, and since the missing leaves are extracted from reality, so to evaluate the point cloud completion capability of MIX-Net on supervised learning, the same experiment was conducted on the seedling leaf datasets in Section 2.2.4, where the missing leaf point clouds were generated, normalized, and then formed a leaf point cloud pair with the complete leaf. The results on supervised learning are given in Table 5 below, and the complete results for missing 50%, 25%, and 15% leaves are given in Figure 9 below.

### 3.6. Nondestructive Leaf Area Measurement Results Using MIX-Net

To verify that our experimental results are helpful in realistic phenotypic measurements, we selected 40 seedlings with occlusion, as shown in Figure 10 below, and then used MIX-Net to isolate the missing leaves of the seedlings, and to complete them. Figure 11 gives the correlation coefficient between the true leaf area (by leaf area meter) and the leaf area before repair with $R^{2}$ = 0.87, MSE = 2.64. The correlation coefficient between the true leaf area (by leaf area meter) and the leaf area after repair after completing is $R^{2}$ = 0.93, MSE = 2.26. The presence of occlusion is particularly serious, we select the leaves which are severely occluded, and from the result of the patching, we can get complete and uniform leaves, and our method provides help for the nondestructive testing of the whole tray of seedlings.

## 4. Discussion

Authors should discuss the results and how they can be interpreted from the perspective of previous studies and of the working hypotheses. The findings and their implications should be discussed in the broadest context possible. Future research directions may also be highlighted.

### 4.1. Point Cloud Classification Results on the Modelnet40 Datasets

The ModelNet40 datasets [43] contain 12,311 CAD models in 40 object classes; it is widely used for point cloud shape classification benchmarks. For a fair comparison, we use 9843 objects from the office for training and 2468 objects for evaluation. The same sampling strategy as PointNet [44] was used, sampling each object uniformly to 1024 points. During training, no data augmentation or voting methods were used during testing. For all network models, the batch size was 32, and each network was trained 250 times individually. The initial learning rate was 0.01, and a cosine function was used to decay the learning rate to adjust the learning rate for each period. The experimental results are listed in Table 6. Compared with PointNet and PCT, MIX-Net improved by 4.2% and 0.2%. The overall accuracy of MIX-Net was 93.4%. It is worth mentioning that our network currently does not consider normal vectors as input.

### 4.2. Point Cloud Segmentation Results on the ShapeNet-Part Dataset

We experimentally evaluate the ShaptNet-Part dataset [46], which contains 16,880 3D models trained to test segmentation from 14,006 to 2874. It has 16 object classes and 50 part labels, and each instance contains no less than two parts. All models were down-sampled to 2048 points, preserving the individual point annotations. The models have a batch size of 16, a training count of 250, and a learning rate of 0.001. Table 7 shows the segmentation results for each type of network. The evaluation metric used is part-average intersection over union. The results show that our MIX-Net improves by 2.0% over PointNet. MIX-Net reaches 85.7%.

### 4.3. Point Cloud Completion Validated on a ShapeNet-Part Dataset

To train our model, we used 13 different objects in the ShapeNet-Part of the benchmark dataset. The total number of shapes is 14,473 (11,705 for training and 2768 for testing). All input point cloud data are centered at the origin, and their coordinates are normalized to [−1, 1]. Ground truth point cloud data were created by sampling 2048 points uniformly on each shape. Incomplete point cloud data were generated by the missing point cloud generation method. We control the parameters to get different numbers of missing points. When comparing our method with other methods, incomplete point clouds with 50% of the original data missing are set up for training and testing. For all network models, the batch size is 16, and each network is trained 100 times separately. The initial learning rate was 0.0001, and a cosine function for learning rate decay was used to adjust the learning rate for each period. Table 8 shows the completion results for each type of network. The evaluation metrics used are CD distance and F-score@1%. The results show that our MIX-Net achieves optimal results in both CD and F-score@1%.

### 4.4. Ablation Experiments

In Table 9 we will discuss the effectiveness of the proposed framework in this paper. We will evaluate it separately in point cloud classification, point cloud segmentation, and point cloud completion. Firstly, for point cloud classification, the feature extraction ability of these modules for point clouds can be verified. Among them, we choose the encoder of PointNet++ as a baseline. Then it is combined with Nas (neighborhood aggregation strategy) and point-mixer to verify the effectiveness of the proposed framework on the modelnet40 datasets. Then for semantic segmentation and instance segmentation of point clouds, we select PointNet++ and ASIS as baselines, respectively, and then experiment on seedling point cloud segmentation datasets. Finally, we use PCN as the benchmark and then validate the effectiveness on leaf restoration on seedling leaf completion datasets. Through the experiments, we demonstrate that our proposed module achieves excellent results in various fields.

## 5. Conclusions

In this study, based on high-throughput data acquisition and deep neural networks, automatic segmentation and completion method for seedling 3D point clouds is proposed. The proposed method can achieve high-quality segmentation and completion from two aspects. Firstly, during the data processing we developed a new method for eliminating hover points and noise points, which can retain more detailed features while removing noise compared with traditional statistical filtering and radius filtering. Secondly, a new network named MIX-Net is proposed to achieve point cloud segmentation and completion simultaneously, which can better balance these two tasks in a more definite and effective way and ensure high performance on these two tasks. Experimental results prove that, compared with state-of-the-art methods, the average performance gain brought by our methods on classification, seedling segmentation, and seedling leaf completion tasks are more than PCT, DGCNN, and Vrc-Net, respectively, leading to more accurate measurement performance on the leaf area phenotypes of seedlings. Furthermore, we also explored the effect of restoration in dealing with the presence of extensive occlusion in the whole tray of seedlings, which provides feasible help for future nondestructive testing of whole-tray seedlings.

## Figures and Tables

**Figure 1 plants-11-03342-f001:**
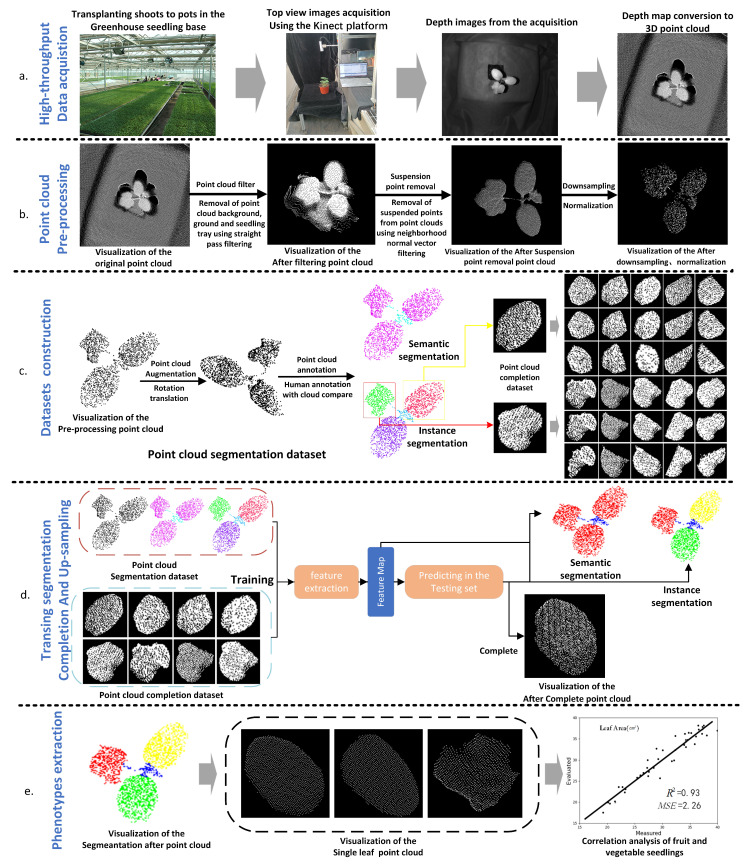
Flowchart of the procedure in this paper. (**a**) High-throughput seeding data acquisition using Kinect. (**b**) Point cloud preprocessing includes point cloud filtering, point cloud down-sampling, and normalization. (**c**) Annotation of data, data enhancement, and missing point cloud datasets construction using Cloud compare software. (**d**) Semantic segmentation of seedlings and missing leaf completion by MIX-Net. (**e**) Phenotype extraction using organ semantic segmentation and missing completion results.

**Figure 2 plants-11-03342-f002:**
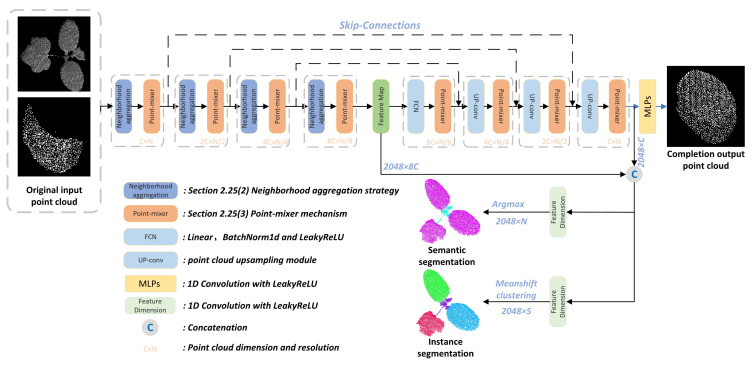
The complete structure of MIX-Net. The **left** part of the figure is the encoder, and the **right** part is the decoder. The neighborhood aggregation module is the neighborhood aggregation strategy proposed in Section 2.2.6. The point-mixer is the multi-level feature fusion module proposed in Section 2.2.7. The UP-conv module is the PointAtrousGraph, the point cloud up-sampling module proposed in the paper.

**Figure 3 plants-11-03342-f003:**
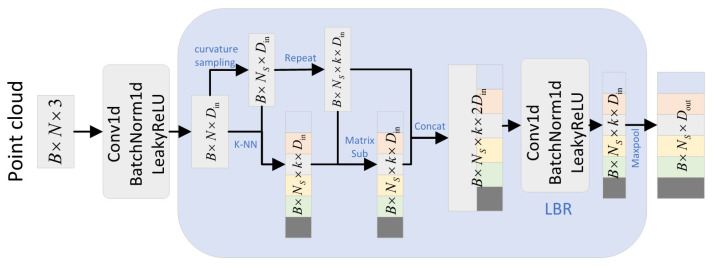
Neighborhood aggregation strategy framework diagram.

**Figure 4 plants-11-03342-f004:**
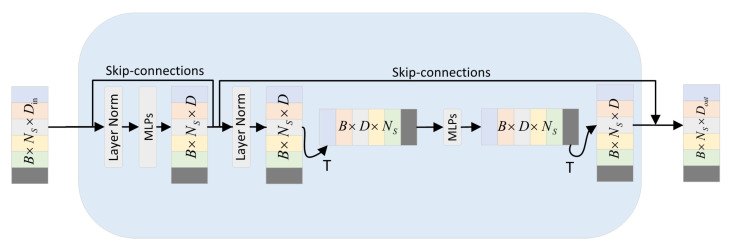
Point-mixer multi-level feature fusion framework.

**Figure 5 plants-11-03342-f005:**
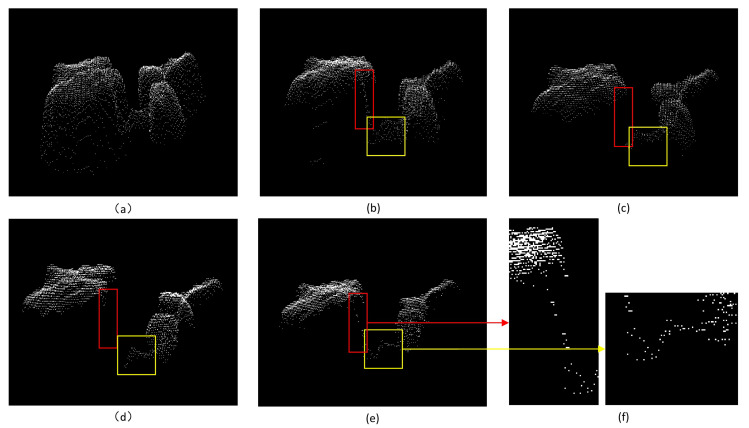
(**a**) The original point cloud with hover point noise. (**b**) Statistical filtering. (**c**) Radius filtering. (**d**) The domain maximum filtering of a two-dimensional depth map. (**e**) Our proposed filtering results. (**f**) denotes the local magnification of our method after filtering.

**Figure 6 plants-11-03342-f006:**
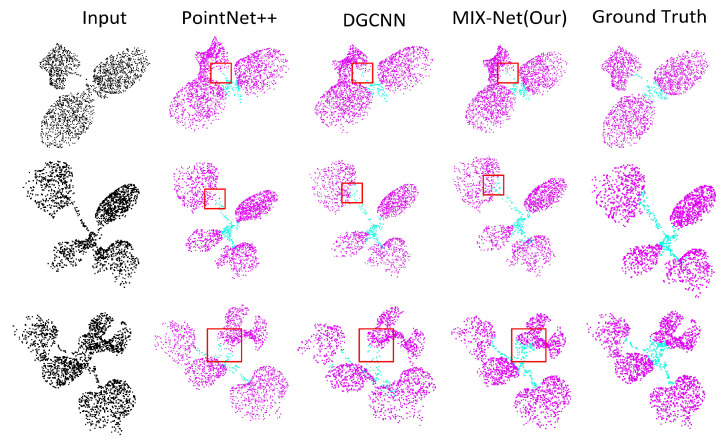
The qualitative semantic segmentation comparison on the three species. DGCNN and PointNet++ are compared with our MIX-Net. The parts with segmentation errors are highlighted by red dotted circles.

**Figure 7 plants-11-03342-f007:**
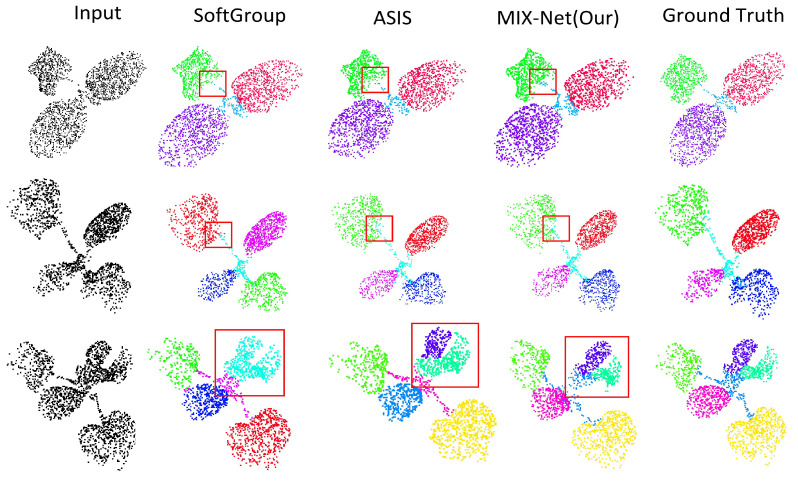
The qualitative instance segmentation comparison on the three species. SoftGroup and ASIS are compared with our MIX-Net. The parts with segmentation errors are highlighted by red dotted circles.

**Figure 8 plants-11-03342-f008:**
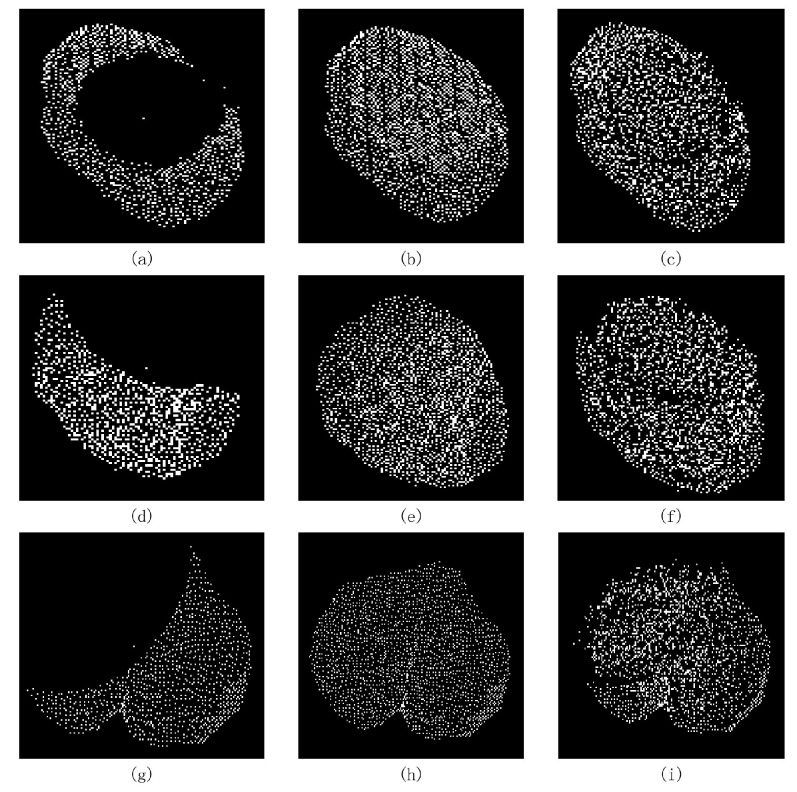
Representation of the complete results of the point cloud leaf datasets on MIX-Net, where (**a**,**d**,**g**) are the missing point cloud leaves from the input network, (**b**,**e**,**h**) are the real complete leaves, and (**c**,**f**,**i**) are the predicted outputs of the MIX-Net network.

**Figure 9 plants-11-03342-f009:**
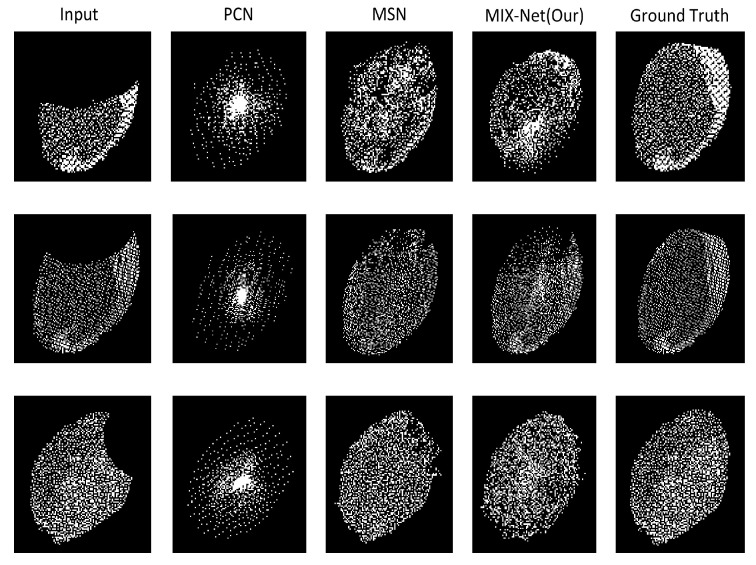
Supervised training results of MIX-Net on the point cloud leaf dataset. In the experiments, the predicted outputs of various methods after supervised training after missing 50%, 25%, and 15%.

**Figure 10 plants-11-03342-f010:**
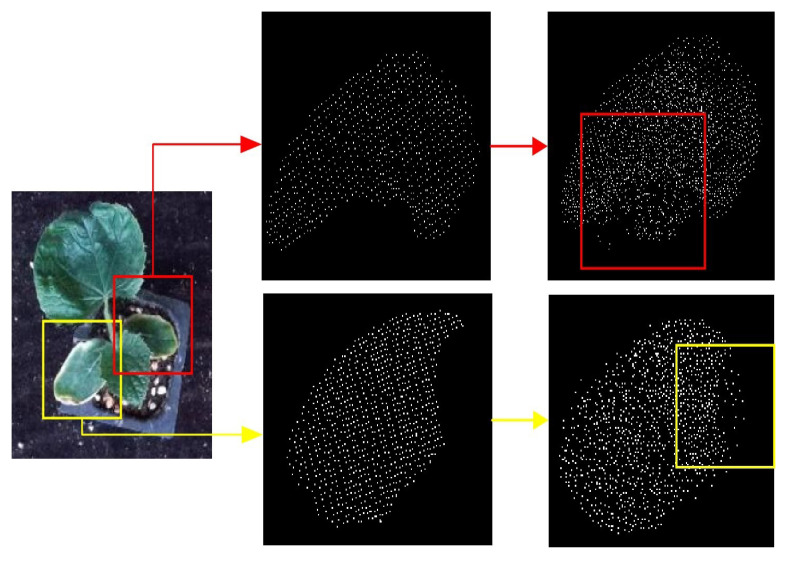
Complementary results of MIX-Net in real phenotypic measurements, where indicates the presence of occlusion in phenotypic measurements of seedlings and our corresponding complementary results.

**Figure 11 plants-11-03342-f011:**
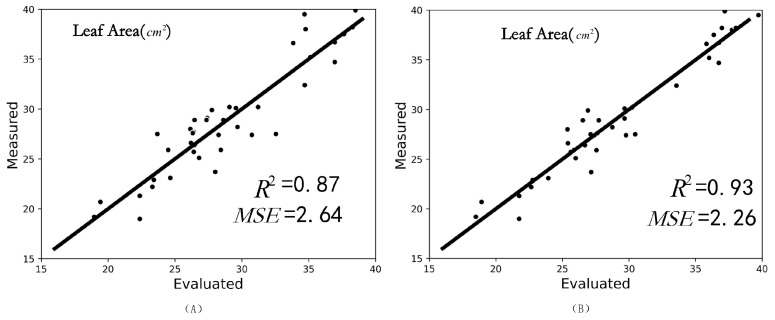
Correlation analysis after leaf area measurement, (**A**) indicates the correlation analysis between the leaf before restoration and the real leaf; (**B**) indicates the correlation analysis between the restored and the real leaf.

**Table 1 plants-11-03342-t001:** Training datasets and testing datasets settings.

	Numder of Training Point Clouds	Number of Testing Point Clouds	Points	Number of Training Point Clouds after Augmention	Number of Testing Point Clouds after Augmention
Number of seedlings point cloud	130	20	2048	520	80
Number of leaf point clouds	500	100	2048	1800	300

**Table 2 plants-11-03342-t002:** Semantic segmentation results of MIX-Net on seedling point cloud datasets.

Methods	Input	Points	mIoU (%)
PointNet++ [24]	P	2048	91.5
DGCNN [21]	P	2048	92.9
MIX-Net (Our)	P	2048	**94.6 **

**Table 3 plants-11-03342-t003:** Instance segmentation results of MIX-Net on seedling point cloud datasets.

Methods	Input	Points	mPrec (%)	mRec (%)
Soft-Group [39]	P	2048	74.26	68.04
ASIS [40]	P	2048	79.13	75.64
MIX-Net (Our)	P	2048	**82.31**	**77.46**

**Table 4 plants-11-03342-t004:** Leaf completion results of MIX-Net on seedling point cloud leaf datasets.

Methods	Input	Points	m-Value	CD × $10^{3}$	EMD
PCN [33]	P	2048	50%, 25%, 15%	1.947	0.106
MSN [41]	P	2048	50%, 25%, 15%	**0.870**	0.072
PF-Net [42]	P	2048	50%, 25%, 15%	1.947	–
Vrc-Net [34]	P	2048	50%, 25%, 15%	1.783	0.107
MIX-Net (Our)	P	2048	50%, 25%, 15%	1.679	**0.071**

**Table 5 plants-11-03342-t005:** The results on supervised learning of MIX-Net on seedling point cloud leaf datasets.

Methods	Input	Points	m-Value	CD × $10^{3}$	EMD
PCN [33]	P	2048	50%, 25%, 15%	1.773	0.113
MSN [41]	P	2048	50%, 25%, 15%	1.914	0.065
MIX-Net (Our)	P	2048	50%, 25%, 15%	**1.276**	**0.063**

**Table 6 plants-11-03342-t006:** Comparison with state-of-the-art methods on the ModelNet40 classification datasets. Precision implies overall accuracy. All cited results are taken from the cited papers. P denotes the number of points, and N denotes the normal.

Methods	Input	Points	Accuracy (%)
PointNet++ [24]	P	1024	90.7
PointNet++ [24]	P, N	1024	91.9
PointCNN [35]	P	1024	92.5
DGCNN [27]	P	1024	92.9
PCT [45]	P	1024	93.2
MIX-Net (Our)	P	1024	**93.4**

**Table 7 plants-11-03342-t007:** Comparison using ShapeNet-Part segmentation dataset. mIoU denotes the average intersection over union. All cited results are taken from the cited papers.

Methods	Input	Points	mIoU (%)
PointNet++ [24]	P	2048	85.1
DGCNN [27]	P	2048	85.2
MIX-Net (Our)	P	2048	**85.7**

**Table 8 plants-11-03342-t008:** Point cloud completion results (CD and F-Score@1%) on the ShapeNet-Part dataset (2048 points).

Methods	Input	Points	m-Value	CD × $10^{3}$	F-Score@1%
PCN [33]	P	2048	50%	2.929	0.29
TopNet [47]	P	2048	50%	3.805	0.38
MSN [41]	P	2048	50%	2.376	0.41
PF-Net [42]	P	2048	50%	3.037	–
Vrc-Net [34]	P	2048	50%	2.881	0.42
MIX-Net (Our)	P	2048	50%	**2.111**	**0.45**

**Table 9 plants-11-03342-t009:** The ablation analysis of MIX-Net. The checkmark stands for the use of a module. The best quantitative values are shown in bold.

classification (modelnet40 dataset)	PointNet++ (encoder) [24]		Nas	Point-mixer	Accuracy (%)	
	✓				90.7	
			✓		89.4	
	✓			✓	92.7	
			✓	✓	**93.4**	
semantic segmentation (seedling semantic segmentation dataset)	PointNet++ (encoder) [24]	PointNet++ (decoder) [24]	Nas	Point-mixer	mIoU (%)	
	✓	✓			91.5	
	✓			✓	92.4	
	✓	✓		✓	93.7	
		✓	✓		91.8	
			✓	✓	**94.6**	
instance segmentation (seedling instance segmentation dataset)	ASIS (encoder) [40]	ASIS (decoder) [40]	Nas	Point-mixer	mPrec (%)	mRec (%)
	✓	✓			79.13	75.64
	✓			✓	77.41	72.36
	✓	✓		✓	81.32	**79.56**
		✓	✓		78.44	76.54
			✓	✓	**82.31**	77.46
Leaf completion (seedling leaf completion dataset)	PCN (encoder) [33]	PCN (decoder) [33]	Nas	Point-mixer	CD × $10^{3}$	EMD
	✓	✓			1.773	0.113
	✓			✓	1.345	0.094
	✓	✓		✓	**1.254**	0.061
		✓	✓		1.493	0.108
			✓	✓	1.276	**0.059**

## Data Availability

The data presented in this study are available on request from the corresponding author.

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
