# Peer review of "MIX-NET: Deep Learning-Based Point Cloud Processing Method for Segmentation and Occlusion Leaf Restoration of Seedlings"

_plants, 2022, doi:10.3390/plants11233342_

Round 1

Reviewer 1 Report

In this manuscript by Han et al., based on high-throughput data acquisition and deep neural networks, automatic segmentation and completion method for seedling 3D point clouds is proposed. The proposed method can achieve high-quality segmentation and completion from two aspects. Firstly, during the data processing, developed a new method for eliminating hover points and noise points, which can retain more detailed features while removing noise compared with the traditional statistical filtering and radius filtering. Secondly, a new network named MIX-Net is proposed to achieve point cloud segmentation and completion simultaneously, which can better balance these two tasks in a more definite and effective way and ensure high performance on these two tasks. Experimental results prove that, compared with state-of-the-art methods, the average performance gain brought by the Han’s methods on classification, seedling segmentation, and seedling leaf completion tasks are more than PCT, DGCNN, and Vrc-Net, respectively. Lead to more accurate measurement performance on the leaf area phenotypes of seedlings. Furthermore, Han et al. explored the effect of restoration in dealing with the presence of extensive occlusion in the whole tray of seedlings, which provides feasible help for future nondestructive testing of whole tray seedlings. I did not actually recognize any real limitations of the manuscript. I think that the current form of the manuscript can be published. However, I have only one suggestion.

Lines 98-112: Please first introduce machine learning and deep learning and mention that Machine learning has been widely used in different fields of plant science such as plant breeding (https://doi.org/10.1016/j.isci.2020.101890), in vitro culture (https://doi.org/10.1007/s00253-020-10888-2), stress phenotyping (https://doi.org/10.1016/j.tplants.2015.10.015), stress physiology (https://doi.org/10.1371/journal.pone.0240427), plant system biology (https://doi.org/10.1007/s00253-022-11963-6), plant identification (https://doi.org/10.1016/j.compag.2016.07.003), and pathogen identification (https://doi.org/10.1094/MPMI-08-18-0221-FI).

Author Response

Dear Editor and reviewers:

Thank you for your letter and the reviewers' comments on our manuscript entitled "MIX-NET: Deep learning-based point cloud processing method for segmentation and occlusion leaf restoration of seedlings" (ID:plants-1994827). Those comments are constructive for revising and improving our paper, as well as the important guiding significance to further research. We have studied the comments carefully and made corrections which we hope meet with approval. The main corrections are in the manuscript, and the responses to the reviewers' comments are as follows (the replies are highlighted in blue).

Replies to the reviewers' comments:

Reviewer #1:

  1. Lines 98-112: Please first introduce machine learning and deep learning and mention that Machine learning has been widely used in different fields of plant science.

Response: Thank you for your valuable comments, we have carefully read the reference papers you listed. And in the introductory section, we have cited the relevant research on deep learning in the field of plants and also added some references you gave.

Once again, thank you very much for your constructive comments and suggestions, which would help us both in English and in depth to improve the quality of the paper.

Thank you and best regards.
Yours sincerely,
Shengyong Xu

Corresponding author:
Name: Shengyong Xu
E-mail: xsy@mail.hzau.edu.cn

Reviewer 2 Report

The study in the manuscript suggested a point cloud processing method based on deep learning for segmentation and occlusion leaf restoration of seedlings. Firstly, a neighborhood space-constrained method to effectively filter out the hover points and outlier noise of the point cloud was proposed to enhance the quality of the point cloud data. Then, a new network named MIX-Net was developed to achieve segmentation and completion of the point cloud. This manuscript has clear logic and complete experiments, but several problems must be considered before publication.

1.    In Figure 1, there is too much-crowded information in the illustration for readers to understand the point work in this study. As a flowchart of the procedure, it is suggested to list only the main parts, and it is unnecessary to show in detail some common parts of such work, such as the annotation of data.

2.    Line 266, the parameters of the point cloud filtering are empirical values. Interpretation of whether there is some relationship between the empirical values and imaging system parameters or experimental material dimensions is needed to confirm the generalizability of this approach.

3.    Line 414, two methods used to be compared with the proposed method seems can not represent the advanced methods in Point-Cloud noise removal. It is suggested to add some SOTA methods to verify the excellent performance of the proposed method.

4.    In Figure 7, the part highlighted by red dotted circles can not clearly express the meaning. It seems that not all the parts that are wrongly segmented are in the circle.

5.    In Figure 9, The scaling ratio seems different between each group of images. For example, g and h.

6.    In Figure 10, red and yellow circles mark opposite regions.

7.    In Table 4 and Line 467. As seen from the data, MIX-Net does not outperform other methods in two metrics, such as MSN being significantly better than MIX-Net in CD*103, and they are performing close in EMD. Taken together it appears that MSN is better. If you want to prove that MIX-Net is more advantageous in the completion of leaf structure, it is advisable to add a comparison with other methods in the Figure 9.

Author Response

Dear Editor and reviewers:

Thank you for your letter and the reviewers' comments on our manuscript entitled "MIX-NET: Deep learning-based point cloud processing method for segmentation and occlusion leaf restoration of seedlings" (ID:plants-1994827). Those comments are constructive for revising and improving our paper, as well as the important guiding significance to further research. We have studied the comments carefully and made corrections which we hope meet with approval. The main corrections are in the manuscript, and the responses to the reviewers' comments are as follows (the replies are highlighted in blue).

Replies to the reviewers' comments:

Reviewer #2:

  1. In Figure 1,there is too much-crowded information in the illustration for readers to understand the point work in this study. As a flowchart of the procedure, it is suggested to list only the main parts, and it is unnecessary to show in detail some common parts of such work, such as the annotation of data.

Response: We have thought deeply about the questions you raised. First of all, what we want to present in Figure 1 is the overall framework of the paper, including data acquisition, data processing, datasets construction, deep network construction, and leaf area measurement. There is a recursive relationship between them, so removing one of them will not be coherent enough. Secondly, in each part of the design there is our own research focus, and we show it together in this part, which can show the completeness of the paper and also save a lot of space. Finally, this is only the general framework of the paper. We have explained the key research sections of the paper at great length later. We hope you can understand this.

  1. Line 266, the parameters of the point cloud filtering are empirical values. Interpretation of whether there is some relationship between the empirical values and imaging system parameters or experimental material dimensions is needed to confirm the generalizability of this approach.

Response: First of all for the setting of the point cloud filtering parameters, there is no relation to the camera parameters as well as to the experimental size. We have conducted experiments not only in the environment of single seedlings, but also in larger plant pots, and the experimental results prove the versatility of our method. We have also given the corresponding annotations in the corresponding section of the text for the doubts you have raised, in order to prevent any further misunderstanding of the same for others.

  1. Line 414, two methods used to be compared with the proposed method seems can not represent the advanced methods in Point-Cloud noise removal. It is suggested to add some SOTA methods to verify the excellent performance of the proposed method.

Response: We have added another point cloud filtering method to the figure for 2D depth maps. This filtering method for 2D depth images is more stable and has good results in point cloud filtering for depth cameras than the previous filtering method for 3D point clouds. We have also obtained better results when comparing with it. If you have other SOTA methods, we hope you will contact us, which will be of great help to our research.

  1. In Figure 7, the part highlighted by red dotted circles can not clearly express the meaning. It seems that not all the parts that are wrongly segmented are in the circle.

Response: We have adjusted the position of the red box to box the most obvious part of the split difference, making the contrast more obvious.

  1. In Figure 9, The scaling ratio seems different between each group of images. For example, g and h.

Response: We have corrected the scale of the corresponding figure, thank you for your revision.

  1. In Figure 10, red and yellow circles mark opposite regions.

Response:Thank you for your attention to detail, which we have adjusted in Figure 10 and which we will pay attention to in future studies.

  1. In Table 4 and Line 467. As seen from the data, MIX-Net does not outperform other methods in two metrics, such as MSN being significantly better than MIX-Net in CD*103, and they are performing close in EMD. Taken together it appears that MSN is better. If you want to prove that MIX-Net is more advantageous in the completion of leaf structure, it is advisable to add a comparison with other methods in the Figure 9.

Response:We would like to provide the following explanations for the doubts you have raised. First of all, in this part of the experiments we performed two different sets of experiments, one of them is the blade repair experiments under self-supervision, and the specific results are given in Table 4 and Figure 8. The second group is the blade repair experiments under supervised learning, and the results are given in Table 5 and Figure 9. The difference between the two groups of experiments is that the experimental data are obtained in different ways, and the data taken by the actual depth camera can only be trained using supervised learning. So in table 5 you can see that our method outperforms pcn and msn across the board. for your comments, we also add the comparison results of other methods in figure 9.

Once again, thank you very much for your constructive comments and suggestions, which would help us both in English and in depth to improve the quality of the paper.

Thank you and best regards.
Yours sincerely,
Shengyong Xu

Corresponding author:
Name: Shengyong Xu
E-mail: xsy@mail.hzau.edu.cn